# Decision Support Frameworks in Public Health Emergencies: A Systematic Review of Dynamic Models in Complex Contexts

**DOI:** 10.3390/ijerph20176685

**Published:** 2023-08-30

**Authors:** Alex S. Príncipe, Aloísio S. N. Filho, Bruna A. S. Machado, Josiane D. V. Barbosa, Ingrid Winkler, Cristiano V. Ferreira

**Affiliations:** 1Oswaldo Cruz Foundation, Rio de Janeiro 21040-360, Brazil; alex.principe@fiocruz.br; 2Department of Management and Industrial Technology, SENAI CIMATEC University Center, Salvador 41650-010, Brazil; aloisio.filho@fieb.org.br (A.S.N.F.); brunam@fieb.org.br (B.A.S.M.); josianedantas@fieb.org.br (J.D.V.B.); 3Institute for Science, Innovation and Technology in Industry 4.0/INCITE INDUSTRIA 4.0, Salvador 41650-010, Brazil; 4Technological Center of Joinville, Federal University of Santa Catarina, Joinville 89219-600, Brazil; cristiano.v.ferreira@ufsc.br

**Keywords:** public health emergency, framework, management, response, prevention, recovery, preparedness, decision making

## Abstract

Public health emergencies are extraordinary events of disease spread, with health, economic, and social consequences, which require coordinated actions by governments and society. This work aims to analyze scopes, application possibilities, challenges, and gaps of decision support frameworks in PHE management, using the components of the Health Emergency and Disaster Risk Management Framework (H-EDRM) and the Preparedness, Prevention, Response and Recovery Model (PPRR Model), providing guidelines for the development of new models. A systematic literature review was carried out using the Web of Science, Scopus, and Pubmed knowledge databases on studies published between 2016 and 2023, and thirty-six articles were selected. The outcomes show a concentration of frameworks on short-term emergency response operations, with a limited emphasis on the political and strategic components that drive actors and responsibilities. Management prioritizes monitoring, evaluation, and information management frameworks. However, the models need to overcome the challenges of multisectoral and interdisciplinary action, different levels of decisions and actors, data sharing, and development of common platforms of evidence for decisions fitted to the various emergencies.

## 1. Introduction

In March 2020, the World Health Organization (WHO) declared the COVID-19 pandemic a Public Health Emergency (PHE) of International Concern affecting the lives of millions of people on a global scale until May 2023, when the slowing of transmission of the coronavirus (SARS-CoV-2) was achieved, but with attention to its consequences. Health emergencies, especially pandemics, epidemics, and disasters, are extraordinary events of disease spread, with health, social, economic, and political consequences, which require coordinated actions by states and affect, especially, the most vulnerable populations [1]. The interaction of a communicable disease (COVID-19) with other non-communicable diseases, such as diabetes and heart problems, affects health systems in the contexts of socioeconomic inequalities and worsens the overall picture, reinforcing the concept of syndemic, when two or more diseases cause amplified damage, according to the living conditions of a population [2].

In emergency situations, the manager makes decisions under dynamic and chaotic conditions, when interpreting the context under pressure from ethical and political constraints [3], seeking to strengthen strategies for the planning, monitoring, and coordination of actions, personnel training, resource allocation, infrastructure and logistics, information management, and risk assessment, priority areas for the development of frameworks, methods, and models for decision support [1]. The challenge may be related to the logistics of allocating respiratory ventilators, a scarce resource in emergencies involving severe acute respiratory syndromes, an action that requires the application of ethical principles, the recognition of the capabilities of health systems, and the limitation of access for vulnerable populations [4].

Considering this complexity, the WHO increased its perspective on response, prevention, recovery, prediction, and preparedness mechanisms to achieve an intelligence hub, with multidisciplinary collaboration, multisectoral decision making, global data sharing architectures, and expanded health context analysis capabilities to support health manager decision making [5].

In public health emergencies, such as the COVID-19 pandemic, each country has the challenge of making decisions tailored to the limitations of its health systems and related subsystems, guided by global guidelines such as the WHO standards, the Sustainable Development Goals (SDGs), the Sendai Framework for Disaster Risk Reduction 2015–2030, and the International Health Regulations [1], among others. These are strategic drivers for positioning states in the face of complexity, in a context conditioned by the behavior of individuals, organizations, and the dynamics of systems, when the gap between the plan and the effectiveness of interventions needs to be managed [6,7].

In this sense, frameworks serve as the interface between the decision maker and the problem to be solved, in various formats such as conceptual models, application of multicriteria methods, and systematic literature review. However, governments can adopt their frameworks without regard for system interoperability and information sharing [8]. In further research, it was found that models focused on specific contexts and with little ability for reformulation of interventions [9], difficulty in incorporating historical data into the analysis [10], and calibration of models for long-term scenarios by the uncertainty levels of an emergency [11].

The bounded rationality of decision makers and the behavior of their heuristics and biases in contexts with variables of uncertainty pose a challenge to framework models [3], requiring them to acquire complicated language information and comprehend the hesitations of decision makers [12]. Consensus production among actors’ nonlinear preferences, dependency on source conditions, and assessments of benefits and losses present a challenge for group decision making [13]. As an input to the decision process, at all decision levels, models must improve the quality of data regarding how managers apply strategies using low or medium certainty evidence [14,15,16].

Country planning for an emergency is comprehensive and must begin before the health event, with the challenges of information analysis, expert decision-making approaches and priority setting, choices of methods and models, budget programming, the establishment of a data infrastructure, and evidence production, which condition alert, preparedness, assessment, and crisis control actions, posing a challenge to managers and government structures [17]; therefore, decision frameworks must be adapted to the country’s government style [18]. Nonetheless, there still need to be more ethical, scientific, and management factors integrated into strategic action planning [19].

The examined works present the challenges of decision frameworks in emergencies, focusing on a gap between model type, actor expectation, and context constraints, which this article seeks to mitigate by describing the characteristics and requirement constraints to support country governments in developing frameworks for emergencies using a more adaptive approach incorporating the system dynamics of [6] and the reality of the context [20].

In this sense, this work aims to analyze the scopes, application possibilities, challenges, and gaps of decision support frameworks in PHE management using components of the Health Emergency and Disaster Risk Management Framework (H-EDRM) [1] and the Preparedness, Prevention, Response and Recovery Model (PPRR Model) [21,22], providing guidelines for the development of new models. The article is organized as follows: Section 2 describes the theoretical background, Section 3 details the materials and methods, and Section 4 describes the results, conclusions, and suggestions for future research.

## 2. Decision Support Frameworks in Public Health Emergencies: Concepts, Complexities, and Managerial Intervention Capacity

In recent decades, the global population has been confronted with recurrent health issues that undermine the ability of governments to respond [23], establishing Public Health Emergencies (PHEs). There are outbreaks, which include an unanticipated growth of a disease in a place, and endemics, which involve infectious agents that are prevalent in a certain geographical area or population. When cases increase in a country or region, the situation is referred to as an epidemic, and state governments proclaim a Public Health Emergency of National Concern (PHENC). The WHO declares a Public Health Emergency of International Concern (PHEIC) when the international scope of an emergency impacts many nations and reaches pandemic proportions [24]. Emergencies emphasize the significance of integrated and multidisciplinary action, considering the One Health concept, which ties human, animal, and ecological health to global health concerns and promotes joint efforts to address a common problem [25].

The WHO guidelines for a public health information center emphasize a predictive perspective to produce better data, better analyses, and better decisions as part of an adaptive, comprehensive, and contextualized approach. Interpreting the context depends on prioritizing risk dynamics rather than managing isolated events, proactive action rather than reactive action, considering the dangers, uncertainties, and vulnerabilities and capacities of each context, as well as involving society and sharing responsibilities. The challenge is directly related to the capacity for resilience in the face of the entropy of the entire health system in an EPS, generated by a scarcity of resources, disputes over political narratives, and indeterminacies between science and common sense [1,5].

As an illustration, the Brazilian government issued an ESPIN in January 2023 due to a lack of assistance to the Yanomami indigenous population, which is characterized by a crisis that requires the mobilization of national strategies in the health, environmental, management, and regulatory dimensions of the territory, in response to the dangers posed by the presence of illegal mining in the Amazon Region [26]. At the global level, the WHO decreed in 2020 the COVID-19 pandemic, a disease caused by the new coronavirus (SARS-CoV-2). This event, along with the H1N1 pandemic (2009), the international spread of polioviruses (2014), the Ebola outbreak in West Africa (2014), the Zika virus (2016), and the Ebola outbreak in Congo (2018) [27], are examples of PHEIC, which, according to the International Health Regulations, are events of risk to the public health of countries and require the involvement of governments and societies [28].

The International Health Regulations, the central instrument for actions to protect countries against the international spread of disease, define public health risk as “the probability that an event will adversely affect the health of human populations, with an emphasis on those that can spread internationally or pose a grave and direct threat” [28]. More than 4500 signals of public health risks are reported annually by the WHO, resulting from natural threats such as earthquakes, cyclones, severe temperatures, and heavy rainfall, environmental degradation, and biological dangers such as airborne and waterborne infections [1].

A PHE is a surprising event generating uncertainty, dealing with turbulent problems, and challenging states’ forecasting and protection strategies. There are resilience challenges to restore the previous equilibrium of a situation, which is not always possible or even enough. These are complex decision-making contexts for managers, requiring collaborative relationships and the strengthening of partnerships at all levels, adoption of innovative solutions, and effective communication about risks to achieve a new reality [29].

Decision making in PHEs requires a complexity calculation as close to reality as possible, considering the information available in the time interval of the decision, considering the scientific, technological, political, and managerial data, when the use of frameworks can mitigate the bounded rationality of the decision makers [30]. This human trait tends to simplify the goal condition while handling complex situations with multiple variables. Frameworks play a role somewhere at the interface between an issue and a decision maker’s heuristics and biases [3].

A framework, based on Design Science Research (DSR), is an “artefact” in the form of constructs with conceptual aspects, models that attempt to describe reality, its variables, and their interactions. In the DSR approach, the method relates steps for the execution of actions and analysis and may adopt heuristics and algorithms; instantiation verifies the application of one or more artifacts in the environment, to know the technical feasibility and effectiveness, according to the characteristics of the context [31]. This concept contributes to the application of frameworks to PHEs when it considers, among other aspects, the need to manage different types of health risks, with specific dimensions, characteristics, and impacts on the relationship of the behavior of disease evolution with the limits of the health systems of the countries [1,5].

One of WHO’s mechanisms for preparing countries for a PHE is the Health Emergency and Disaster Risk Management Framework (H-EDRM), an expert-developed management framework that reinforces the paradigm of an integrated approach to deal with health emergencies and disaster risks [1], guiding states in Preparedness, Prevention, Response and Recovery measures (PPRR Model) [21,22] for intervention plans. Figure 1 presents the actions required for a country’s response and the H-EDRM components, dimensions that articulate and guide the development of frameworks, according to the contexts of decision making in PHEs.

The prevention measures reduce the impact of hazards and risk events on health, especially in the more vulnerable population. Recovery is a slow process that promotes actions to restore community life, such as physical, social, or psychological rehabilitation. The response measures include urgent life-saving support, such as the deployment of workforce, infrastructure, logistics, and communication systems. Preparedness increases PHE readiness by strengthening the capacity of governments, institutions, and society through proper legislation, resource planning, and community empowerment [21,22].

The H-EDRM Policies, Strategies, and Legislation (H-EDRM-PSL) outline organizational structures, roles, and duties. Planning and Coordination (H-EDRM-P&C) emphasize the integration of health planning at all levels. The Monitoring and Evaluation (H-EDRM-M&E) component tracks risks and the implementation of strategies, programs, and activities. The Information and Knowledge Management (H-EDRM-I&KM) integrates health surveillance, early warning, and evidence-based technical guidance. Health Infrastructure and Logistics (H-EDRM-HI&L) address safe and sustainable health facilities. Human Resources (H-EDRM-HR) build capacity and establish human competencies to address PHEs. The Health and Related Services (H-EDRM-H&RS) component guides actions such as prevention, basic care, rehabilitation, and immunization programs. The Risk Communication (H-EDRM-RC) component reports communication strategies to society, and the Community Health Capacities (H-EDRM-CHC) component assesses community risk prevention, population engagement, and territory planning. Finally, the Financial Resources (H-EDRM-FR) mobilize program budgets, contingency funds, and financial arrangements [1].

As shown in Figure 1, the relationship between intervention measures and management components provides a comprehensive framework for decision making, assessing, and implementing PHE activities. Within a plan with porous borders and interconnected action cycles, it is possible to find an integrated perspective of numerous activities to minimize vulnerabilities and strengthen national plans.

Effective decision making in public health emergencies integrates institutions, public policy, science, and society; integrates multidisciplinary and multisectoral collaboration platforms; bridges knowledge generation and action; overcomes fragmentation; and strengthens the sharing of comprehensive and contextualized data, thereby empowering decision makers to understand and manage health risks. Reliable decisions adopt analytical methods and tools, access collective knowledge, and include cognitive characteristics of the decision maker [5]. The complexity of decision making in conducting programs and projects in a PHE full of uncertainty [29] integrates the behaviors of individuals and organizations, connectivity, dependencies, system dynamics, and context ambiguities [7].

A PHE is a complex process guided by the interdependence of political, social, economic, and health issues, which involves local, regional, and global components [20]. In this way, the framework for decision making in a PHE is developed as a complex adaptive system, which are nonlinear systems conditioned by their initial contexts and agents’ interactions [32], categorizing the dynamics of structures and behaviors [6].

## 3. Materials and Methods

This systematic literature review is a qualitative approach, which allows the identification of key issues in the field [33]. Our study on decision support frameworks in contexts of public health emergencies is exploratory because it is complex and challenges governments and society preparedness. Thus, it is useful to the researcher to investigate variables of this context’s complexity which are not yet known [33].

This review followed the Preferred Reporting Items for Systematic Reviews and Meta-Analyses (PRISMA) guidelines, which aims to “help systematic reviewers transparently report why the review was done, what the authors did, and what they found” [34]. This review is structured into planning, scoping, searching for published research, assessing the evidence base, synthesis, analysis, and writing stages [35]. It has an Open Science Framework registration, https://osf.io/u6xfb (accessed on 15 June 2023). The strategy resulting from this validation process is described in the following sections.

Studies on decision making in PHEs are recent, and a bibliometric analysis indicates an increase since 2014, associated with the intensification of health events, with 73.27% of the publications identified in the last 5 years, with an indication of growth in the coming years. The results indicate that there is little coordination between countries, institutions, and authors about decision making in PHEs, which raises the possibility of interdisciplinary analysis from a multicultural perspective which is more interpretative of the context of problems that affect the management dimension, especially in countries with populations in situations of economic, social, and environmental vulnerability that intensify in a PHE [36]. In another study on frameworks for evaluating responses to epidemics in low-income countries, the authors highlight the importance of a planned systematic review on the subject, as there are still gaps in the configuration of response mechanisms, with limited structures for evaluating the life cycle of interventions according to the characteristics of the context, without prioritizing iterative processes and the capacity to redesign interventions [9].

This article reports on the general aspects of frameworks. It considers complexity through the interface of individuals and organizations, dependent on the connectivity of systems and the ambiguities and uncertainties of contexts [7]. The intention is to understand the need for frameworks to adapt to different methods of government, local realities, organizational cultures, and societal perspectives and to contribute to studies of decision making in PHEs in a more comprehensive approach of a complex adaptive system, with nonlinear relationships and multiple agents who interpret contexts from different perspectives [37].

In this sense, our article covers the last 7 years in a systematic literature review on aspects and configurations of decision support frameworks in PHEs, which interprets the interface of management problems through the components of the H-EDRM [1] and the PPRR Model [21,22], guided by questions to promote an interdisciplinary analysis of the issues identified, as follows in the planning.

### 3.1. Planning

Planning defines the scientific knowledge bases that will be investigated in the research [35] as follows: Web of Science, Scopus, and Pubmed. They were selected because they are reputable, multidisciplinary, and worldwide scientific databases with extensive citation indexing coverage and provide the finest scientific publishing data. Scopus includes 87 million curated documents [38]. Web of Science covers more than 87 million records [39], and Pubmed over 35 million citations [40].

### 3.2. Defining the Scope

In defining the scope, guiding questions were formulated to be addressed to achieve the objective of this study. The text is organized by the H-EDRM and the PPRR model, which address specific topics. The questions guide the analysis of the frameworks studied. These are the following: Q1: How are management decision support frameworks characterized in PHE contexts? Q2: What type of intervention are the frameworks geared towards in a PHE? Q3: What are the challenges and gaps to be overcome by frameworks for strengthening decision making in PHE contexts?

Considering the importance of an integrated view and the boundaries between PPRR measures and the H-EDRM components in PHEs, the identification of the theme should be in more than one H-EDRM, which could accommodate an interdisciplinary approach. However, we tried to locate the article studied in the H-EDRM with the greatest intensity, without prejudice to bringing them closer together in the composition of the texts and analyses.

### 3.3. Bibliographic Research

In the literature search step, following the questions presented in the scoping, a string is used to query the databases [35]. In these databases, the Title, Abstract, and Keywords fields were searched for publications published between 2016 and 2023, recent years of health emergencies, such as the Zika epidemic (2014–2016), the Wild Yellow Fever epidemic (2017), and the COVID-19 pandemic (2020) [41].

The definition of the search string in the knowledge bases analyzed bibliographies linked to the topic and accessed the Health Sciences Descriptors (DeCS) base (https://decs.bvsalud.org/sobre-o-decs/ accessed on 16 June 2023) to find words and synonyms, resulting in the following string composition: (“public health”) OR (“health”) OR (“Public Health Administration”) AND (“humanitarian”) OR (emergenc*) OR (“emergency management”) OR (“disaster”) OR (“outbreak”) AND (“evaluation”) OR (“assessment”) OR (“appraisal”) AND (framework*) OR (“structure”) OR (“method”) OR (“model”) OR (“tool”) OR (“mechanism”) OR (“technique”) AND (“decision-making”) OR (“Decision Support Techniques”) NOT (“medical”). The search eliminates medical research that has no relation to the areas of management and administration.

The study plan investigated the management of PHE response, prevention, recovery, and preparedness, as well as its synonyms and derivations. The description examines the PHE, as well as its synonyms and derivations; the management, administration, and evaluation frameworks; and the frameworks implemented, which can be mechanisms of different types, approaches, and applications of methodologies and models. The knowledge bases’ definition of the search term searched bibliographies utilizing keywords from articles on the subject and the DeCS/MeSH multilingual thesaurus.

### 3.4. Database Evaluation

In the evaluation stage, the inclusion and exclusion criteria are defined to select the documents that will be relevant for this research, according to the questions chosen in the scope definition [35].

To perform the search for studies related to the research topic, the filters pre-established by the respective databases were prioritized. In the Web of Science, these included Health Care Sciences Services, Medical Informatics, Environmental Sciences, Medicine General Internal, Computer Science, Artificial Intelligence, Emergency Medicine, Multidisciplinary Sciences, Management, Health Policy Services, Nursing, Computer Science Interdisciplinary Applications, Infectious Diseases, Pharmacology Pharmacy, Medicine Research Experimental, Water Resources, Computer Science Information Systems, Environmental Studies, Psychiatry, Social Sciences Interdisciplinary, Operations Research Management Science, Tropical Medicine. In Scopus, these included Social Sciences, Computer Science, Mathematics, Health Professions, Decision Sciences, Business, Management and Accounting, Economics, Econometrics and Finance. For Pubmed, all studies were looked at. After this step of applying the filters, 715 studies were identified.

The selection resulted in the deletion of 661 items after reading the Title, Abstract, and Keywords fields, leaving 54 titles, which, when combined with the suppression of duplicates, produced a total of 32 studies. Then, 4 additional titles from other sources of interest were included, bringing the total number of studies selected for analysis to 36, categorized according to the following exclusion criteria:E1: Exclude studies that do not address frameworks for health-related decision making.E2: Exclude specialized framework studies irrelevant to management from the fields of epidemiology, biomedicine, and clinical area decisions.E3: Exclude non-English-written studies, due to the impossibility of evaluating titles in all languages and English being the predominant language in the knowledge bases searched.E4: Exclude research published before 2016.

### 3.5. Synthesis and Analysis

Figure 2 illustrates the systematic flow in the PRISMA model with the 36 selected studies.

PRISMA only presents the studies chosen to analyze the frameworks. The other references cited are used as a theoretical basis for the terms mentioned in the article. These studies were read and analyzed, seeking to understand the relationships between the results presented to identify patterns, divergences, and research opportunities and to answer the proposed guiding questions. The 36 studies identified after performing the described procedures are summarized in Table 1, identifying the H-EDRM, the PPRR model, and the related theme, which will be analyzed in Section 1. 

In total, 24 titles out of the total number of studies included in this study fall between 0.7 and 2.15 on the Journal Citation Indicator (JCI), with 1.0 being the global average [42]. Notably, the H-EDRM WHO Framework [1] does not include JCI, but it is a significant document of guidelines since its components contain categories that will be used to arrange the study described in this article, as shown in Table 1.

## 4. Results

This section presents the analysis of the selected frameworks, according to the appendix available at https://bit.ly/3QZ6DUA (accessed on 15 June 2023). The results are organized by the H-EDRM and answer the presented questions. Q1 addresses the characterization of the frameworks: contexts, identification of the H-EDRM and the PPRR model, scope, and method or model of analyses. Q2 identifies the type of intervention by H-EDRM components and PPRR model measures and the interface between them. Q3 analyzes the gaps and challenges in an interdisciplinary approach. The dynamics observed in our framework study are those located between the causes of a health emergency with its ambiguities and uncertainties and the limits of health systems, whose contexts present challenges for management, based on which we seek to identify their dynamics and sense of intervention, scopes, and relationship of variables and tendencies.

### 4.1. Frameworks Dynamics: Scope, Intervention, and Gaps

The frameworks are presented in several formats, adopting conceptual structures [10,16,17,20]; the use of methods and modeling with a multicriteria approach [11,12,13,43,44,45,46,47,48,49,50,51,52,53,54,55,56,57,58]; the application of questionnaires with experts and managers working in PHEs, [14,19,23]; and the description of steps and guidelines for guiding the context analysis [1,4,8,9,15,18,59].

Figure 3 presents a set of characteristics of the studies evaluated in this article, observing the objects and formats of the frameworks studied, grouped according to the components of the H-EDRM [1].

Figure 3 shows the strategic plan for managing PHEs and addresses policies, planning, and coordination related to studies concerned with the resilience of structures [20,47], action plans [16,17], and government strategies [18], as well as ethical values [19] that challenge the positioning of nations in the face of a context of uncertainty, particularly for vulnerable populations.

The dimensions of information and monitoring and evaluation of the results of managers’ responses deal with the evolution of risk dynamics that need to be managed in emergencies [10,23,45,52,55,59]. This is a challenge related to the production of evidence to support the best decisions at all levels [12,13,46,57] but with challenges of its own generation and quality [14,15]. Information is an input to the decision-making process, and its sharing between states and health agencies needs to be prioritized [8] so that the response is adapted to the reality of the context [9], which implies knowing the government’s strategies [43,56] and the WHO’s guidelines [58].

The provision of structures and resources focuses on hospitals that are under immediate pressure in PHE situations. The focus is on the agility of adapting physical infrastructure [48] and levels of health security [51], technology allocation [4], and ideal stocks of health supplies [49]. Regarding human resources, studies have focused on the perception of human error in contexts of uncertainty and stress on health professionals [3,60].

A PHE requires the attention of different areas according to the ambiguities and uncertainties of a context, considering a multidisciplinary approach that includes economic, social, psychological, and health impacts [5], which can be observed in the authors’ areas of knowledge, such as in the Operation Management Area, Indian Institute of Management Ranchi [51], the School of Economics and Finance, Queensland University of Technology [60], and the Centre for Complexity Science, University of Warwick, Coventry, UK [53]. Figure 3 presents a set of characteristics of the studies evaluated in this article, observing the objects and formats of the frameworks studied, grouped according to the components of H-EDRM [1]. In another analysis, the organization of frameworks by authors and countries, as shown in Figure 4, presents a predominance of Chinese, Canadian, British, and American authors.

Identifying frameworks using H-EDRM and PPRR Models implies observing permeable borders on their components to look at management dynamics in the context of a PHE. In this sense, Figure 5 shows the frameworks studied associated with an element of the most incredible intensity.

Due to the magnitude that the COVID-19 pandemic has reached since 2020, response efforts predominate, followed by preparedness, prevention, and a recovery measure, suggesting a focus on the system’s reaction; however, the strengthening of preparedness activities affects the evolution of all health systems during a crisis, resulting in a more effective response and capacity for resilience to recovery of the health system.

Future research must investigate the conditionality relationship between these actions to determine the extent to which having more preparedness actions can produce more assertive responses and require less time to recover structures and the relationship between measure types and management components, as depicted in Figure 5. In addition, the challenge of decision support frameworks in emergency situations is to increase the attention to the life cycle of a PHE, integrating the management of preparedness, preventive, recovery, and response aspects and overcoming a step-by-step approach [9].

Framework development incorporates decision support concepts, theories, and multicriteria methods, which address the “multidimensional representation of problems with identification of critical information, comparisons of alternatives, and preference structures” [61]. Although this study of decision support frameworks in PHEs does not intend to exhaust the analysis of the limits and shortcomings of multicriteria methods, which would imply considering that synergistic relationships should be strengthened between multicriteria analysis, cognitive psychology, decision-maker behavior, and decision support systems, our work identifies the methods and their applications cited in some studies, as shown in Table 2, whose dynamics will be addressed by H-EDRM as follows.

#### 4.1.1. Policies, Strategies, and Legislation | H-EDRM-PSL

The H-EDRM-PSL component presents The Analytical Framework of China’s Governance Responses to COVID-19 (response), with centralized governance for effective coordination of implemented measures, in a framework that integrates institutional, administrative, coordination, and governance configurations, with rational behavior of the actors, which generate a dynamic to reduce positive entropy in management that compromised government responses during the pandemic [18]. Decision support frameworks in PHEs must consider context-specific characteristics, governance models, and governance structures [20], as in the Chinese model, which prioritizes centralized actions, adjusting national strategies to individuals, called agents [56].

However, as the frameworks established by nations reflect the opinions and circumstances of their authors, the exchange of models between different country realities is not automatic, and it is vital to expand PHE research into the operational and ethical dimensions of models that address the dynamic complexity of management in a context [20]. It is also important to note that “one size fits all” approach strategies present governments with the difficulty of determining the spatial and temporal effects of different measures in relation to the characteristics of populations within the same country or territory, particularly those with greater economic and social vulnerability [56].

What must be investigated is the suitability of frameworks in diverse PHE situations, with models that include characteristics of the nations’ governance style in accordance with the degree of centralization of activities. The absence of command and the perception of a PHE as a political game compromises the responses of governments, as in the United States during the COVID-19 pandemic; however, the Chinese experience of more effective control of the pandemic is not easily adopted by other nations because their decision-making contexts differ [18].

#### 4.1.2. Planning and Coordination | H-EDRM-P&C

H-EDRM-P&C addresses mechanisms for planning intervention in an epidemic. The definition of action plans is presented in a framework to support the integration of priority setting in the preparedness, alert, control, and evaluation stages of a disease pandemic, using documentary analysis and incorporating the experiences of health professionals, in an outbreak of the Ebola virus in Uganda [17]. In this study, there is a conceptual framework for developing health emergency preparedness plans using the Health Technology Assessment (HTA) approach, which includes health problem and technology use, safety, clinical effectiveness, cost and economic evaluation, ethical analysis, organizational aspects, and social and legal aspects, but also missing a 360° perspective on interventions in PHE settings, with evidence and best practices, considering different stakeholders in the decision-making process [16].

The countermeasure strategies present a multicriteria decision support system for choosing strategies to mitigate (prevention) the effects of COVID-19 using Bayesian networks to addresses uncertainty in dynamic environments, in the transition from short term to long term. Despite the improvement in multicriteria decision support methods for health emergencies, the scoring adjustment of the attributes of analysis may be harmed in contexts of uncertainty, for dealing with strategies not previously tested and for more uncertain attributes that receive less weight in the analyses, compromising a broad view of the problem [53].

In terms of resilience, managing COVID-19 presented a hybrid approach that identifies causal factors, security barriers, and key lessons using bowtie modelling. Modelling societal resilience in health emergencies integrating causation, barriers, and lessons learned relies on hypothetical scenarios rather than probabilities and immediate problem-solving that is not always possible. Few studies have focused on frameworks for dynamically addressing the context of uncertainty, able to integrate accumulated management experiences from previous health crises to understand how contexts returned *(recovery)* to a state of normality [47]. The resilience framework for PHE preparedness is another example of the interactions of a complex adaptive system for resilient public health. It is a qualitative study with health professionals in Canada which integrated the dynamic approach of the context’s components of ethics, governance, leadership, feedback loops, nonlinearity, self-organization, adaptability, interconnectedness, and risk analysis to promote resilience. The challenge is to consider in the analysis that frameworks developed by countries must reflect their circumstances; this means that the capacity for resilience of a country is related to its preparedness, which conditions the way of response and prevention and recovery measures [20].

Ethics, as a component of the planning and coordination measures in a PHE, also depends on developing ethically robust policies for community protection. The framework of ethical values for planning, policy enactment, and action on disasters, based on a decision guide, includes benefit and harm, effectiveness, precautionary principles, harm from intervention, respect, equity, and trust. The framework is a decision tree that provides guidance on the potential of the crisis, the establishment of recommendations, communications for intervention, and the verification of results, but the achievement of an adequate level of protection (prevention) for the population in public health emergencies, based on evidence, still lacks models that integrate the scientific, ethical, and management dimensions, with social, infrastructure, and logistical factors, for interventions that are appropriate to the context, especially for the least favored populations [19].

In problem analysis, planning for a PHE is a complicated system with interdependent variables [20]. Thus, Multicriteria Decision Support Methods (MCDMs) are used in multidimensional problem representation, the identification of critical information, comparisons of alternatives, and preference structures, being most effective in a combined approach [43,50,58]. Despite the importance of mathematical models for scenario projection, the calibration of probabilities is more reliable for short time frames (two weeks) and is degraded over the long term according to the amount of trajectory uncertainty, intervention strategy, and decision quality in a PHE [11].

#### 4.1.3. Human Resources | H-EDRM-HR

H-EDRM-HR components are strategic resources to face pandemics because they reinforce guidelines for personnel planning, occupational health, personnel safety, and capacity building in technical, epidemiological, diagnostic, service, and communication areas, in all cycles of the PPRR model and even for the H-EDRM components, at all levels [1]. Despite this importance, only two frameworks were identified in our study. There was a human error causal category structure applied for the systematic retrospective analysis of incidents in high-risk domains that includes external and organizational influences, unsafe supervision, and preconditions for unsafe acts (prevention). Considering the need for a more comprehensive approach that understands the human error of the health professional involved in dealing with a PHE at all levels, the model was adapted to include the sociopolitical context, ecological influences, the national regulatory framework, the international regulatory framework, the social environment, and acts of sabotage. Health emergencies occur in turbulent contexts like COVID-19, with redoubled efforts of health professionals and other sectors, which are influenced by psychological, organizational, social, and regulatory frameworks. Despite the interest in knowing these factors, studies of causal chain frameworks to mitigate human error still are topics for future research [60].

The framework for training public health practitioners in crisis decision making was also identified as a necessity for the *response* to a PHE to straighten the decision-making process in complex emergency contexts, guided by the domains of decision theory, ethical principles, and the individual and group emotions of decision makers. The tool was based on the OODA loop (Observe, Orient, Decide, Act), which incorporates intuition and deliberative process in choosing alternatives. Chaos and unpredictability condition decision making in emergencies, establishing a gap between the intuitions, heuristics, and biases of the decision maker. What is missing is a structured analysis of the variables of the context of uncertainty to achieve a more rational and deliberative process. It is a challenge to be studied to guide the training processes of health authorities in this context. Human resources can also be understood in the face of a PHE by the integration of agents participating in the decision-making process [3].

The alignment of different agents in PHE decision making requires overcoming collective risk framing positioning, nonlinear preferences, dependence on source conditions, and estimation of gains and losses of each actor or group of actors. Additionally, cognitive and complex linguistic information capture presents a challenge. The gaps point to the development of consensus models which integrate psychological factors, information quality, and decision speed. Thus, decision-maker hesitation and perception demand weights and ranking of alternatives in structured analysis to mitigate their bounded rationality [12,13] and achieve an adequate deliberative process through a better integration of heuristics, biases, and analysis of multiple variables [3]. It is difficult to establish comprehensive, systematic, and standardized evaluation indexes in real situations [54], when organizations and governments, individuals, or groups act according to their own policies, resulting in the compromised interoperability of systems and knowledge generation on common challenges, such as cross-border disease transmission [8].

#### 4.1.4. Monitoring and Evaluation | H-EDRM-M&E

In H-EDRM-M&E, the analysis of risk dynamics in contexts of uncertainty requires the development of decision support frameworks with the application of advanced hybrid dynamic methodologies to overcome the shortcomings of conventional probabilistic static models. The framework evaluates risk dynamics in time-dependent contexts and the dynamic variation of process variables, using a dynamic risk analysis (DRA) for mapping, managing, and identifying risk dynamics [52], with the possibility of application in health preparedness measures. In another study, a nomenclature framework for assessing disaster-related health risk was used to evaluate accuracy, historical evidence, uncertainties of context, hazard, its impact and exposure, vulnerability and capacity assessment, and risk characterization. The challenge to be overcome by the analysis frameworks is the development of assessment models with reproducible and validated processes, capable of incorporating available historical data or their absence for prevention and predictions, especially in new events or those extremely rare ones that generate uncertainties—the black swan [10].

Governments strategies are subject to evaluating factors to respond to the COVID-19 pandemic, taking care to apply the guidance factors established by the WHO for countries. The framework uses Decision-Making Trial and Evaluation Laboratory (dematel) and Fuzzy Rule-Based Techniques and identifies causal relationships between factors and critical elements impacting Covid-19 virus infection, like social media sharing, mass crowding control, people movement control, international travel restrictions, distance learning, and economic stimulus measures. However, the study calls attention to the challenge of linking multicriteria decision support methods to the Dematel method to analyze complementary components (factors) influencing the spread of COVID-19, including different statistical techniques with larger sample sizes [58]. The evaluation of the national effectiveness of strategies implemented by governments to address COVID-19 with TOPSIS q-rung orthopair fuzzy method and its association establishes alternatives in extended contexts, with a lack of information, ambiguities, uncertainties, and dissensions among experts. The analysis includes mandatory quarantine, restriction of movement, herd immunity, and early diagnostic testing [43].

The adjusting government strategies to contain SARS-CoV-2 transmission by analyzing the interactions and behaviors of agents integrates geographical factors, climatic factors, displacements, and contacts between agents in public spaces, called residents in quarantine or confirmed cases, the transportation agents, or even the government agent which establishes policies to address the emergency. The model relates the agents, their attributes, and interactions, based on data for simulation in the time and space of the analysis, in multi-agent modeling, which uses the tool Repast Simphony 2.8.1 (Argonne National Laboratory, Lemont, IL, USA), developed in the Eclipse Platform and using the Java language. It analyzes the spatial dynamics and produces, from scientific evidence, dynamic feedback about the interactions in time, allowing the government to analyze scenarios and adjust prevention and control actions, adopting effective intervention measures according to the context. The restrictive measures for the population are incorporated according to the behavior of the agents and the dynamics of the contexts, without the prevalence of a one-size-fits-all approach. However, this creates a challenge for governments in knowing the impacts of different measures in space and time, given by the economic and social development of a region [56].

Particularly regarding the humanitarian response in PHE contexts, the identification and assessment of factors facilitating agility capacity in the aid network, the framework includes the timeliness of delivery of products and services, health risk management, long-term needs, health information and donor organization, and government and non-governmental organization policies to face the challenge of overcoming the gaps in verifying the effectiveness of action plans (response). It is an application of a hybrid evaluation approach that combines fuzzy decision trial evaluation laboratory (DEMATEL) and the analytic network process (ANP) to deal with ambiguities, uncertainties, and inaccuracies inherent in the evaluation process. The methods assess interactions between elements and establish reliability ratings [44].

Monitoring and evaluating disaster actions within the scope of the health department (responses) is a framework that generates records for learning and preparing future responses to address emergencies. It is a conceptual framework developed based on a literature review on quality improvement, and it can be applied to enhance the disaster monitoring and evaluation cycle, including stages of documentation, evaluation, dissemination, and implementation of measures, looking at context settings and response mechanisms, focusing on the studies that explain ways of intervention and operation in different contexts [59].

Monitoring and evaluating risk dynamics [10,52], governments strategies [43,58], humanitarian response [44], and disaster [59], means overcoming the gaps in verifying the effectiveness of action plans and living with uncertainties and ambiguities that impact on the vulnerabilities, capacities, and levels of exposure to risk experienced by the populations in a PHE [10], increasing the entropy of the health systems, thus creating a dynamic of variables and responses that require performance monitoring. The evaluation of the performance of an epidemic includes the establishment of real-time evolution and forecasts, trying to verify the assertiveness of interventions with metrics of calibration, sharpness, and biases of the predictions. It is a study of the Ebola epidemic (response) in Sierra Leone (2014), a model combining a mechanistic core (Susceptible–Exposed–Infectious–Recovered, SEIR model) with non-mechanistic variables, observing time series and some level of uncertainty about underlying processes. Despite the importance of mathematical models for decision making and the projection of scenarios in health crises, the calibration of probabilities is more reliable for short periods (two weeks) and is compromised in the long term, depending on the level of uncertainty of the trajectory, the intervention strategies, and the quality of the recommended decisions [11]. In addition to analysis using the Technique for Order Performance by Similarity to Ideal Solution (TOPSIS) to rank alternatives according to the distances between positive and negative ideal solutions, Vise Kriterijumsa Optimizacija I Kompromisno Resenje (VIKOR) to order and select alternatives with conflicting criteria, and the Complex Proportional Assessment (COPRAS) to rank alternatives according to importance and utility, future research should address methods able to generate the optimal solution further from the non-ideal measures which managers face in a PHE [45].

It is known that the performance of health systems in a PHE allows for an adaptive response for epidemics, which addresses the analysis of context, intervention, process, performance, and impact, and a route to performance analysis and optimization of interventions, but the studies show models focused on specific contexts, with low capacity to reformulate the intervention, frameworks with limited structures for reapplication in different contexts and an iterative approach, missing settings of evaluation of the full cycle of an intervention [9].

#### 4.1.5. Information and Knowledge Management | H-EDRM-I&KM

The H-EDRM-I&KM brings together frameworks to support better evidence for decision making, first on an application by the Emergency Operational Committee (EOC) of the Saskatchewan Health Authority, Canada, to identify COVID-19 policy priorities, providing rapid reviews and evidence reporting, in a web-based data dashboard. The COVID-19 evidence support team produce and sustain the best available evidence on COVID-19 to facilitate the decision making of policymakers, researchers, and clinical practitioners and foster the context-based learning cycle in health, which depends on information sharing; however, an aspect still to be overcome is the performance of experts in isolated silos of their own characteristics, which compromises the integration of knowledge from multiple actors and the generation of quality evidence in the emergency [14]. Another study highlights that the quality of evidence can impact positively or negatively on the effect of decisions, the communication strategies of governments, and the definition of prospective plans; however, studies are needed on how leaders deal with evidence of low or medium certainty in health emergencies. It is the Grading Of Recommendations Assessment, Development And Evaluation (Grade), the product of an open community (www.gradeworkinggroup.org—accessed on 15 June 2023), that checks and makes recommendations on the quality of evidence to support decision making, applied in the context of public health emergencies [15]. 

A disaster impacts the environment and public health, and it is understood that it requires going through different phases of risk mitigation (preparedness, prevention, recovery and response). However, risk assessment frameworks need to advance the identification of evidence in the early stages of the emergency (response) with the involvement of experts and specialized and multidisciplinary technical contributions. In this way, a methodology was developed to strengthen decision making, communication, planning, and action management to assess risks, an on-line tool that integrates the European Union Civil Protection Mechanism [23]. In another study, a National Risk Analysis Model (NRAM) used in the COVID-19 pandemic scenario projection integrates expert knowledge with real-word facts to provide a broader context. The model incorporates, among other factors, the area affected by the disease, outbreak management and health (prevention) measures, population density, standard of living, youth population distribution, and government efficiency. Its structure used the Bayesian Network (BN), with the interdependence of factors, in association with the Interpretive Structural Modeling (ISM) method and the K2 algorithm, to establish a network analysis structure [55].

The decision-making process to response and prevention plans to the COVID-19 pandemic, according to the studies, requires the application of multicriteria methods for decision support, integrating the expression of uncertainty by decision makers, setting criteria weights for choosing an optimal solution [46], and ranking alternatives based on the quality of available information and on psychological factors of the decision maker [12,13], also the values for a range or set of variables and not exact values [57]. The challenges are the application methods restricted to experts and decision makers and their linguistic expressions, without considering information from different sources and characteristics, which may cause biased results of response ranking for emergencies [46], and the incorporation of the dynamics of the psychological process of decision makers into decision support frameworks on risk propensity or aversion in pandemic processes [57]. The dimensioning of weights for gains and losses of decisions in health emergencies involves a cognitive dimension, capturing complex linguistic information, and understanding the hesitations of decision makers. However, the production of a consensus still needs to be prioritized as an element of collective positioning on vulnerabilities and uncertainties of the context [12]. Consensus production in group decision-making conditions the effectiveness of responses in contexts of uncertainty, since it deals with the positioning of each decision maker as to the framing of risks, nonlinear preferences, dependence on origin conditions, and the estimation of gains and losses. What remain as aspects to be overcome are the relationship between the models of consensus production, the psychological factors of decision makers, and the quality of available information and the speed of decisions required for prevention and control actions in health emergency [13].

The decision-making process asks for a system of emergency intelligence with the capability to deal with a context full of uncertainty and complexity. In this way, studies have been carried out about interagency decision support systems for sharing critical information in cross-border situations, which addresses the emergency management life cycle, integrating preparedness measures with stages of planning, simulations, and training; response measures using a decision support system and real-time monitoring; and recovery measures for vulnerable groups, highlighting the interoperability of existing systems and end-user-oriented solutions [8]. In another study, a multicriteria decision support method composition of Probabilistic Uncertain Language Term Set (PULTS) with the VIKOR method was developed to evaluate the intelligence capability in large public health event emergencies in probabilistic uncertain language environments; however, few studies address the topic due to the difficulty of establishing comprehensive, systematic, and standardized evaluation indexes of the context’s real situation [54]. 

#### 4.1.6. Health Infrastructure and Logistics | H-EDRM-HI&L

H-EDRM-HI&L presents framework studies on safety risk factors in the hospital environment during COVID-19 in India, using a Fuzzy Delphi Method (FDM). The weights of categories and their respective factors are calculated and ranked based on their criticality using the Fuzzy Analytic Hierarchy Process (FAHP), and the ranking of Indian hospitals is conducted by the Fuzzy Technique for Order of Preference by Similarity to Ideal Solution (FTOPSIS). The preparation of health services for the response to a pandemic integrates guidelines from reference bodies such as the WHO and can adopt multicriteria decision support methods to analyze risk factors. However, the exact definition of the criteria and the attribution of values for analysis of services is a challenge to be overcome due to the dynamics of contexts, ambiguities of expert opinions, and scarcity of resources that affect hospitals mainly during the PHE [51].

A literature review on health care agility to classify hospitals in Tehran, Iran, in the context of disaster and the application of the Flowsort method with IT2FSs (interval type-2 fuzzy sets) build a framework to classify hospitals by level of agility, based on management capacity to deal with disasters in contexts of uncertainty. It verifies the relationships between agility factors and the actions to prepare, mitigate, rebuild, and respond to a disaster situation, including organizational structure, hospital chain management, clinical governance, technological upgrading, market awareness, and trained staff. Hospital agility is a structuring component of health service responses in situations of natural or man-made disasters in terms of how effectively facilities respond to the emergency, with competence, flexibility, and speed [48]. A pandemic in its magnitude affects countries and occupies the limits of physical structures and health services, which means the readjustment of facilities for new uses. The availability of frameworks for assessing the flexibility and resilience of health facilities is scarce for models that minimally address structural, organizational, and technological aspects of the spaces. Hospitals are very pressured structures in PHE contexts, as they suffer from a shortage of human and material resources and quickly need the adaptation of structures, service protocols, and necessary technological innovations, which requires the flexibility analysis of health facilities, addressing hospital structure and façade, and building a plan considering expandability, restrictions, interchangeability, and functionality [62].

COVID-19 exposed the weaknesses of supply chain systems by the sudden increase in global demand for health supplies, which challenged the limits of production and logistics and increased the need for studies on frameworks for assessing optimal stock levels in preparedness measures for health emergency contexts. In this way, a decision model about stock levels of health systems in emergency scenarios was developed, using multicriteria spherical fuzzy regret. It represents the level of hesitation or adherence of decision makers to listed alternatives, considering the availability of local production and proximity to production centers, necessary quantity of product or equipment, supply costs, the criticality of use, inventory, and stock maintenance [49].

The spread and speed of cases of a severe acute respiratory syndrome may increase the need for equipment and inputs such as mechanical ventilators and oxygen. A conceptual framework for determining the need for and allocation of ventilators during a PHE was developed (response), capable of planning actions and adequate logistics of equipment, supplies, and qualified personnel in intensive care, analyzing patient demand and the need for adequacy and increase in resources in hospitals, strengthening provisions at federal, state, and local levels. Despite the definition of strategies for technology allocation, frameworks for the situation analysis of the health system need to contemplate the capacity of structures, qualified personnel, and especially ethical principles to serve different population groups [4].

#### 4.1.7. Health and Related Services | H-EDRM-H&RS

H-EDRM-H&RS listed only one study in decision support methods for managers on improvements for the reduction in overcrowding in emergency rooms in Paraná, Brazil, which addresses feedback relationships, interdependence, and influences among criteria, with emphasis on resource utilization, layout efficiency, productivity and technology, in a composition of Decision-Making Trial and Evaluation Laboratory—DEMATEL—and Preference Ranking Organization—PROMETHEE II. The management of the health emergency room to avoid overcrowding deals with problems that are difficult to identify and with patient management factors that cannot be changed quickly; however, according to the complexity of the context, there is a need for the association of multicriteria decision support methods to prioritize alternatives, such as assigning weights, establishing causal relationships between criteria, identifying improvement, and ordering the choices in time [50].

To the best of our knowledge, the studies did not present a specific financial resources framework (H-EDRM-FR); however, it is a strategic resource to provide financing expeditiously and maintain all the programs and measures in different stages of a PHE. The lack of studies also applies to risk communication (H-EDRM-RC), which plays a special role in positioning society in complex contexts of contradictory information that compromises risk mitigation strategies and the protection of citizens, especially vulnerable populations. Communication is closely related to the level of response according to a society’s community capacities (H-EDRM-CHC), which includes health professionals and civil society in shared crisis planning with the community [1], but no studies were identified in this area either. The human resources component (H-EDRM-HR), which determines the allocation of skilled employees for the efficacy of PHE initiatives, did not emerge as a priority in the research studied. In terms of infrastructure and logistics (H-EDRM-HI&L), the frameworks emphasize the hospital environment, but the issue could be broadened to include sanitation, transportation, and energy factors that affect the health system, as well as more studies on the supply chain, with a focus on preparedness frameworks for future emergencies.

## 5. Conclusions

Governments and societies face challenges for assertive decision making in contexts of health emergencies, where the complexity of management lies between the health event’s behavior and the health system’s limit, implying a multidisciplinary and interdisciplinary, multisectoral, and contextualized approach. In this sense, the framework is at the interface between the decision maker and the problem, which in a context of uncertainty takes on nonlinear and counterintuitive proportions.

The analysis of the frameworks using H-EDRM recorded the prevalence of components in the areas of information and knowledge, monitoring and evaluation, planning and coordination, and infrastructure and logistics; however, policies, strategies and legislation, human resources and services, risk communication, and the community capacity were of a lower priority. In the analysis, response measures predominate, followed by preparedness, prevention, and recovery. This conclusion presents a question for future research, which is the conditionality relationship between measure types and management priorities in a PHE, to verify to what extent strengthening the preparedness action can generate more assertive responses and a shorter recovery time of health structures.

This article does not exhaust the analysis of the gaps in multicriteria methods, as synergistic relationships must be strengthened between multicriteria analysis, cognitive psychology, decision-maker behavior, and decision support systems [61] for a more dynamic approach to the complexity of the decision context, which may be the subject of future research.

In conclusion and based on the analysis of aspects regarding context constraints, the historical perspective of emergency behavior and future impacts, the limits of provision of technologies and infrastructure, governance styles, and decision maker cognition factors, this article presents the following guidelines for the development of decision support frameworks in PHEs:Adapt the development of frameworks to the country’s governance model, according to its organizational forms, vulnerabilities, and capacities, to mitigate the introduction of models that are disconnected from reality.Prioritize ethical aspects in the planning of PHEs in all actions (preparedness, response, prevention, and recovery), with special attention to the most vulnerable populations, incorporating historical data to know the behavior of society in previous events and strengthen the resilience capacity for future emergencies.Strengthen policies, strategies, and legislation to prepare the states, prioritizing planning and scenario projection as a conditioning step for response time, improving the effectiveness of interventions and the recovery of health systems.Integrate the expectations and hesitations of decision makers at all levels, individual and collective, with frameworks that best interface heuristics and bias and the rationale of context in the face of variables of uncertainty.Develop within the framework functions that incorporate community capacity into PHE decision making to produce appropriate responses at all levels and increase the social resilience through collective action to strengthen the capacity of the society to deal with the effects of positive entropy in complex and dynamic PHE contexts.Apply methods capable of incorporating the linguistic expressions of the agents, feedback relations, and interdependence among criteria in framework development, making hypotheses and behaviors explicit in the context analysis, due to the uncertainties that compromise the accuracy of predictions.Improve evidence sharing and analysis frameworks, with quality analysis and guidance on levels of certainty, to define local strategies aligned to global challenges in PHEs. In this regard, the interoperability of information systems needs to be strengthened.Forecast complexity scenarios in all phases of a PHE, continuously confronting the behavior of the health event and the limits of the health services and the supply chain, production, and logistics to develop strategies for allocating strategic inputs to health, including human, cognitive, financial, and technological resources.Expand the capacity of the communication frameworks with society, as a strategic action to avoid misinformation and to strengthen sensemaking in society about the interventions.

The studies have shown that the main challenge for the development of frameworks is supporting an adaptive outlook by the manager regarding the complexity of PHE contexts, with tools and mechanisms suited to different circumstances, models of government, and society, since the intervention takes place in a part of a larger reality.

## Figures and Tables

**Figure 1 ijerph-20-06685-f001:**
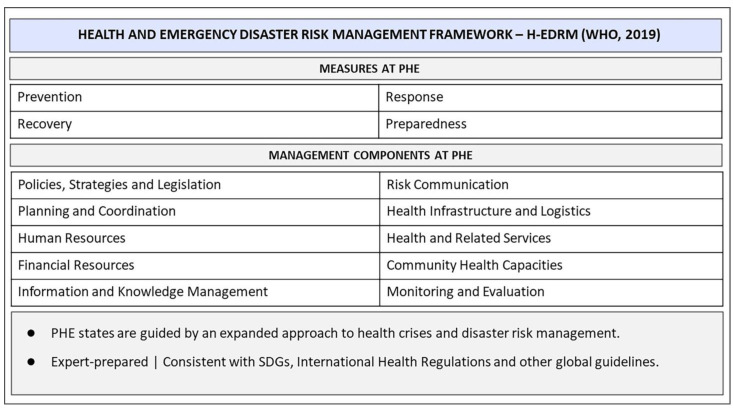
H-EDRM—measures and components— Source: Own authorship.

**Figure 2 ijerph-20-06685-f002:**
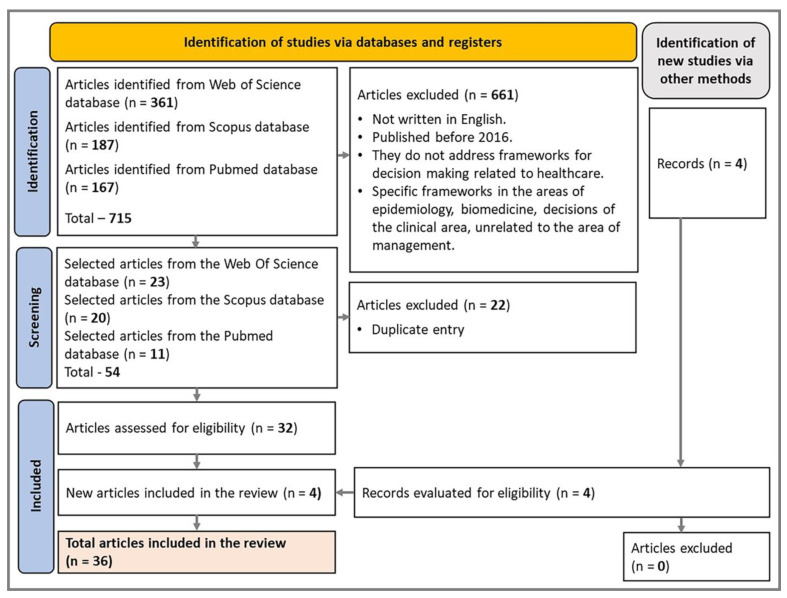
Systematic review flow diagram, adapted from PRISMA [34].

**Figure 3 ijerph-20-06685-f003:**
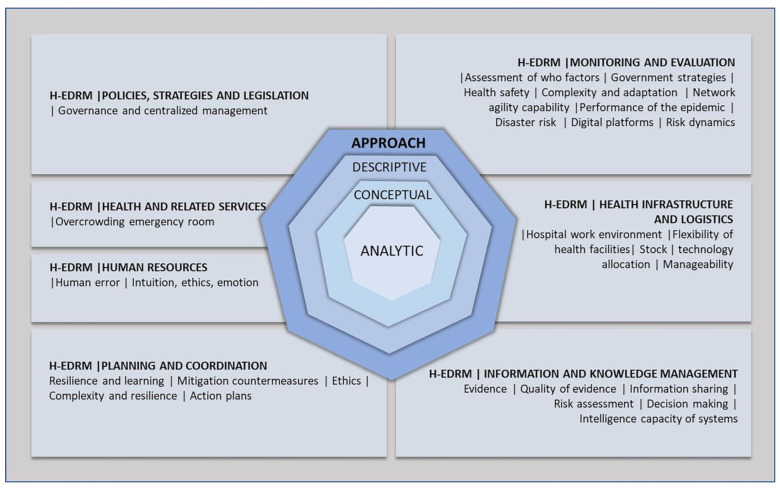
Frameworks: scopes by H-EDRM.

**Figure 4 ijerph-20-06685-f004:**
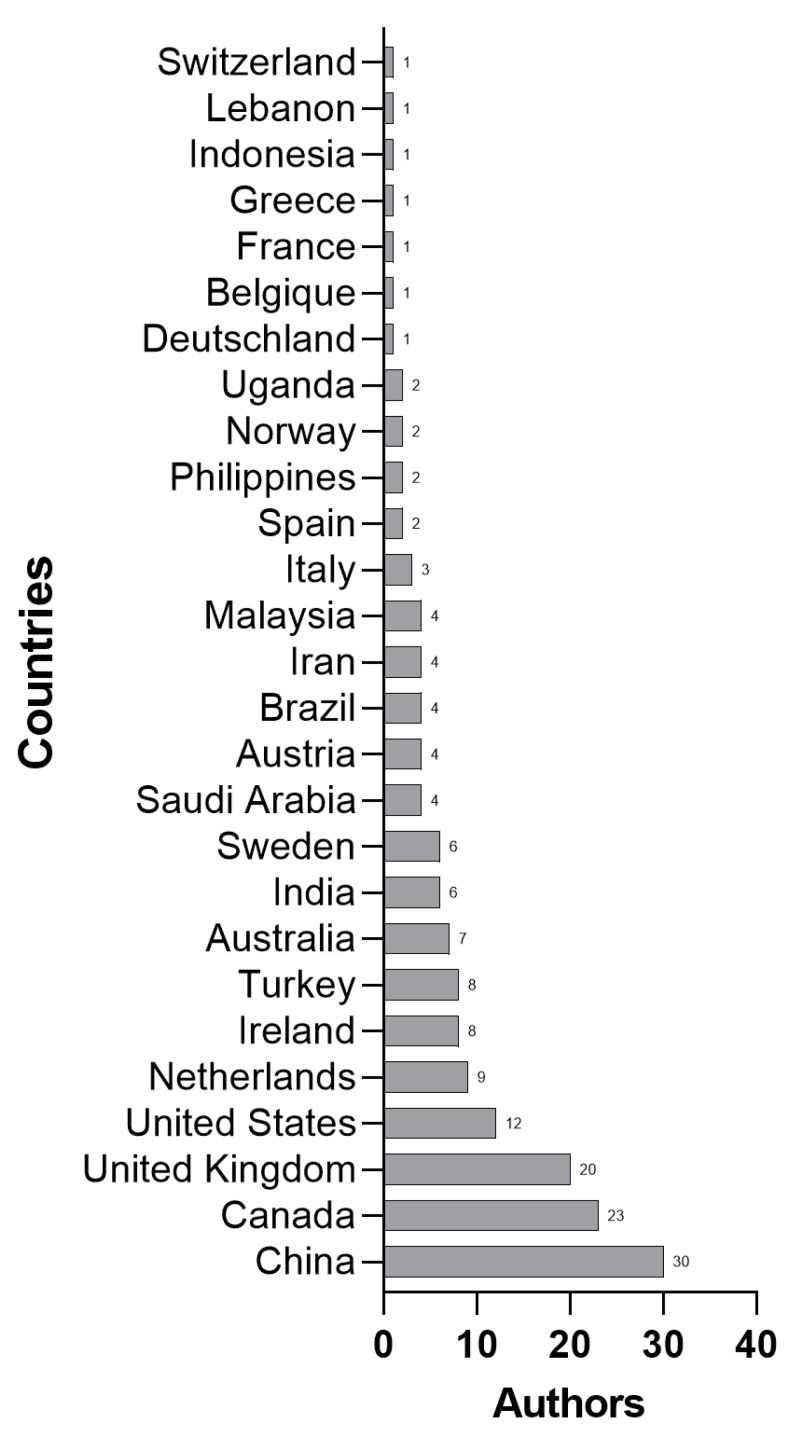
Frameworks: authors by country.

**Figure 5 ijerph-20-06685-f005:**
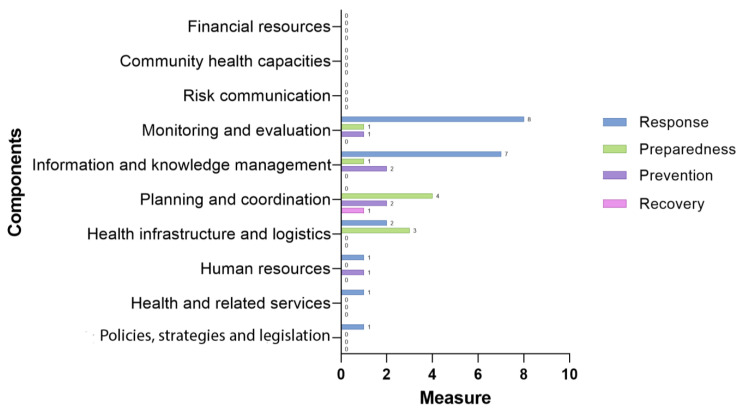
Frameworks by H-EDRM components and measures.

**Table 1 ijerph-20-06685-t001:** Studies identified in the systematic literature review. https://bit.ly/3QZ6DUA (accessed on 15 June 2023).

Authors	Year	JCI	Selected Studies—Title	H-EDRM	PPRR Mode	Subject
(Shi et al., 2022)	2022	2.15	An extended multi-attributive border approximation area comparison method for emergency decision making with complex linguistic information	H-EDRM-I&KM	Response	Decision-making method
(Schünemann et al., 2020)	2020	1.69	Using GRADE in situations of emergencies and urgencies: certainty in evidence and recommendations matters during the COVID-19 pandemic, now more than ever and no matter what	H-EDRM-I&KM	Response	Quality of evidence
(Warsame et al., 2020)	2020	1.58	Towards systematic evaluation of epidemic responses during humanitarian crises: a scoping review of existing public health evaluation frameworks	H-EDRM-M&E	Response	Adaptive approach to epidemic intervention cycle
(Alkan & Kahraman, 2021)	2021	1.57	Evaluation of government strategies against COVID-19 pandemic using q-rung orthopair fuzzy TOPSIS method	H-EDRM-M&E	Response	Government strategies
(Pegoraro et al., 2020)	2020	1.56	A hybrid model to support decision making in emergency department management	H-EDRM-H&RS	Response	Reduction of overcrowding
(Hezer et al., 2021)	2021	1.3	Comparative analysis of TOPSIS, VIKOR and COPRAS methods for the COVID-19 Regional Safety Assessment	H-EDRM-M&E	Response	Assessing health security
(Deng et al., 2023)	2023	1.29	A national risk analysis model (NRAM) for the assessment of COVID-19 epidemic	H-EDRM-I&KM	Prevention	Risk analysis
(Bickley & Torgler, 2021)	2021	1.16	A systematic approach to public health-Novel application of the human factors analysis and classification system to public health and COVID-19	H-EDRM-HR	Prevention	Human error
(Labib, 2021)	2021	1.16	Towards a new approach for managing pandemics: Hybrid resilience and bowtie modelling	H-EDRM-P&C	Recovery	Resilience
(McDonald et al., 2020)	2020	1.11	Facemask use for community protection from air pollution disasters: An ethical overview and framework to guide agency decision making	H-EDRM-P&C	Prevention	Ethical values
(Funk et al., 2019)	2019	1.11	Assessing the performance of real-time epidemic forecasts: A case study of Ebola in the Western Area region of Sierra Leone, 2014–2015	H-EDRM-M&E	Response	Evaluating performance of an epidemic
(Jing, 2021)	2021	1.05	Seeking opportunities from crisis? China’s governance responses to the COVID-19 pandemic	H-EDRM-PSL	Response	Government strategies
(Khan et al., 2018)	2018	1.02	Public health emergency preparedness: a framework to promote resilience	H-EDRM-P&C	Preparedness	Resilience
(Asadi et al., 2022)	2022	1.01	Evaluation of Factors to Respond to the COVID-19 Pandemic Using DEMATEL and Fuzzy Rule-Based Techniques	H-EDRM-M&E	Response	Assessment of the WHO guiding factors
(Goode et al., 2021)	2021	1	Development of a Rapid Risk and Impact Assessment Tool to Enhance Response to Environmental Emergencies in the Early Stages of a Disaster: A Tool Developed by the European Multiple Environmental Threats Emergency NETwork (EMETNET) Project	H-EDRM-I&KM	Response	Risk assessing
(Huang et al., 2022)	2022	0.93	New method for emergency decision making with an integrated regret theory-EDAS method in 2-tuple spherical linguistic environment	H-EDRM-I&KM	Response	Decision-making method
(Brambilla et al., 2021)	2021	0.93	Flexibility during the COVID-19 Pandemic Response: Healthcare Facility Assessment Tools for Resilient Evaluation	H-EDRM-HI&L	Preparedness	Health facilities—hospitals
(Keim, 2018)	2018	0.89	Assessing Disaster-Related Health Risk: Appraisal for Prevention	H-EDRM-M&E	Prevention	Assessing disaster-related health risk
(Groot et al., 2021)	2021	0.88	Developing a rapid evidence response to COVID-19: The collaborative approach of Saskatchewan, Canada	H-EDRM-I&KM	Response	Producing evidence
(Gossip et al., 2017)	2017	0.83	Monitoring and evaluation of disaster response efforts undertaken by local health departments: a rapid realist review	H-EDRM-M&E	Response	Monitoring and evaluating the disaster cycle
(Kapiriri et al., 2021)	2021	0.82	A framework to support the integration of priority setting in the preparedness, alert, control and evaluation stages of a disease pandemic	H-EDRM-P&C	Preparedness	Action Plan
(Kayman & Logar, 2016)	2016	0.81	A Framework for Training Public Health Practitioners in Crisis Decision-Making	H-EDRM-HR	Response	Learning for decision making
(Moheimani et al., 2021)	2021	0.8	Assessing the agility of hospitals in disaster management: application of interval type-2 fuzzy Flowsort inference system	H-EDRM-HI&L	Preparedness	Classifying hospitals by level of agility
(Yang & Guo, 2022)	2022	0.73	Evaluation of Emergency Intelligence Capability of Major Public Health Events in Probabilistic Uncertain Language Environment	H-EDRM-I&KM	Preparedness	Emergency intelligence systems
(Pan et al., 2022)	2022	0.67	Assessment model for rapid suppression of SARS-CoV-2 transmission under government control	H-EDRM-M&E	Response	Government strategies
(Strong et al., 2021)	2021	0.67	Building a Bayesian decision support system for evaluating COVID-19 countermeasure strategies	H-EDRM-P&C	Prevention	Countermeasure strategies
(Koonin et al., 2020)	2020	0.66	Strategies to Inform Allocation of Stockpiled Ventilators to Healthcare Facilities During a Pandemic	H-EDRM-HI&L	Response	Allocation of technology
(Raveendran et al., 2022)	2022	0.59	A comprehensive review on dynamic risk analysis methodologies	H-EDRM-M&E	Preparedness	Dynamic risk analysis
(Miglietta et al., 2021)	2021	0.48	Health technology assessment applied to emergency preparedness: a new perspective	H-EDRM-P&C	Preparedness	Action Plan
(Lv et al., 2022)	2022	0.43	A group emergency decision-making method for epidemic prevention and control based on probabilistic hesitant fuzzy prospect set considering quality of information	H-EDRM-I&KM	Prevention	Group decision-making method
(Wang, 2022)	2022	0.42	Grey multiattribute emergency decision-making method for public health emergencies based on cumulative prospect theory	H-EDRM-I&KM	Response	Decision-making method
(Dashtpeyma & Ghodsi, 2021)	2021	0.42	Humanitarian relief chain agility: identification and evaluation of enabling factors	H-EDRM-M&E	Response	Agility capacity of humanitarian aid network
(Onar et al., 2020)	2020	0.32	Multi-criteria spherical fuzzy regret-based evaluation of healthcare equipment stocks	H-EDRM-HI&L	Preparedness	Stock levels of health systems in emergency
(Rathore & Gupta, 2022)	2022	0.3	A fuzzy based hybrid decision-making framework to examine the safety risk factors of healthcare workers during COVID-19 outbreak	H-EDRM-HI&L	Response	Safety risk factors in hospital
(Neville et al., 2016)	2016	0.29	Towards the development of a decision support system for multi-agency decision-making during cross-border emergencies	H-EDRM-I&KM	Response	Sharing critical information
(WHO, 2019)	2019	NI	Health Emergency and Disaster Risk Management Framework	H-EDRM-P&C	Preparedness	WHO strategies

**Table 2 ijerph-20-06685-t002:** Methods and applications in the evaluated studies. https://bit.ly/3QZ6DUA (accessed on 15 June 2023).

Authors	Year	JCI	H-EDRM	PPRR Mode	Multicriteria Method	Application in the Evaluated Study
(Shi et al., 2022)	2022	2.15	H-EDRM-I&KM	Response	Composition: DHHLTSs, MABAC	Incorporate the decision maker’s uncertain information into the evaluation of alternatives, weighting of choice criteria, and ranking of solutions.
(Alkan & Kahraman, 2021)	2021	1.57	H-EDRM-M&E	Response	TOPSIS q-rung orthopair fuzzy method.	Establish alternatives in extended contexts, with a lack of information, and ambiguities, uncertainties, and dissensions among experts.
(Pegoraro et al., 2020)	2020	1.56	H-EDRM-H&RS	Response	Composition: DEMATEL, PROMETHEE II	Address feedback relationships, interdependence, and influences among criteria.
(Hezer et al., 2021)	2021	1.3	H-EDRM-M&E	Response	Comparison: TOPSIS, VIKOR, COPRAS	Ranking alternatives according to the distances between positive and negative ideal solutions (TOPSIS). Sort and select alternatives with conflicting criteria (VIKOR). Ranking alternatives according to importance and utility (COPRAS).
(Deng et al., 2023)	2023	1.29	H-EDRM-I&KM	Prevention	Composition: Bayesian Network (BN), ISM, K2 algorithm.	Establish networks with expert knowledge and case study data from context. Identify network factors and parameters for scenario production and sensitivity analysis.
(Labib, 2021)	2021	1.16	H-EDRM-P&C	Recovery	Resilience modeling and bowtie modeling.	Analyze the interaction between a temporal and causal axis.
(Asadi et al., 2022)	2022	1.01	H-EDRM-M&E	Response	Composition: DEMATEL, Fuzzy	Identify causal relationships between critical factors and elements that impact virus infection.
(Huang et al., 2022)	2022	0.93	H-EDRM-I&KM	Response	Composition: 2-Tuple spherical linguistic term set, CRITIC, EDAS	Address expressions of uncertainty by decision makers, rank alternatives, and establish criteria weights for an optimal solution.
(Moheimani et al., 2021)	2021	0.8	H-EDRM-HI&L	Preparedness	Composition: Literature review; Flowsort with IT2FSs	Incorporate agility factors and linguistic expressions into the analysis. Establish alternatives based on sorted criteria and categories.
(Yang & Guo, 2022)	2022	0.73	H-EDRM-I&KM	Preparedness	Composition: PULTS, VIKOR	Assess intelligence capability in PHE, uncertain probabilistic language environments.
(Strong et al., 2021)	2021	0.67	H-EDRM-P&C	Revention	Bayesian networks	Address uncertainty in dynamic environments in the transition from short term to long term.
(Raveendran et al., 2022)	2022	0.59	H-EDRM-M&E	Preparedness	Dynamic risk analysis (DRA)	Identify the dynamics of risk.
(Lv et al., 2022)	2022	0.43	H-EDRM-I&KM	Prevention	Fuzzy (PHFS). Cumulative Prospect Theory (CPT)	Analyze probabilities of performance of the decision object, integrating decision makers’ choices in a risk context.
(Wang, 2022)	2022	0.42	H-EDRM-I&KM	Response	Application of the Cumulative Prospect Theory (TCP).	Weighting functions in contexts of uncertainty and risk for decision gains and losses. Establish the possibility of values for a range or set of variables, rather than exact values.
(Dashtpeyma & Ghodsi, 2021)	2021	0.42	H-EDRM-M&E	Response	Composition: DEMATEL, ANP.	Evaluate interactions between elements and establish consistency ratings.
(Onar et al., 2020)	2020	0.32	H-EDRM-HI&L	Preparedness	Multicriteria spherical fuzzy regret.	Represent the level of hesitation or adherence of decision makers to listed alternatives.
(Rathore & Gupta, 2022)	2022	0.3	H-EDRM-HI&L	Response	Composition: Fuzzy FDM, FAHP, FTOPSIS.	Identify risk factors. Calculate and rank weights of categories and factors according to their importance. Classify hospitals.

## Data Availability

The data presented in this study is fully available in the Appendix A. The file is available at https://bit.ly/3QZ6DUA (accessed on 15 June 2023).

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
