# Peer review of "Decision Support Frameworks in Public Health Emergencies: A Systematic Review of Dynamic Models in Complex Contexts"

_ijerph, 2023, doi:10.3390/ijerph20176685_

Round 1

Reviewer 1 Report

 line 54 Gave WHO acronym but continues to write out World Health Organization.

Figure 1 should be landscape. Very difficult to read when it is in such small font. 

Author Response

Dear Reviewer, 

We are pleased to resubmit the manuscript of our paper entitled Decision Support Frameworks in Public Health Emergencies: A Systematic Review of Dynamic Models in Complex Contexts. We appreciate your recommendations, which considerably contributed to the improvement of the work. Please find a detailed description of the changes made below:

Point 1 - line 54 Gave WHO acronym but continues to write out World Health Organization.

Response 1 - We appreciate the feedback. The acronym WHO - World Health Organization has been revised in the text, as well the following acronyms, in order to improve the flow of the text: Public Health Emergencies (PHE), Health Emergency and Disaster Risk Management Framework (H-EDRM); H-EDRM Policies, Strategies, and Legislation (H-EDRM-PSL), Monitoring and Evaluation (H-EDRM-M&E), Information and Knowledge Management (H-EDRM-I&KM), Health Infrastructure and Logistics (H-EDRM-HI&L), Human Resources (H-EDRM-HR), Health and Related Services (H-EDRM-H&RS), Risk Communication (H-EDRM-RC), Community Health Capacities (H-EDRM-CHC), Financial Resources (H-EDRM-FR), and Preparedness, Prevention, Response and Recovery Model (PPRR Model).

Point 2 - Figure 1 should be landscape. Very difficult to read when it is in such small font.

Response 2 - The picture has been altered and inserted into the text.  According to the general changes suggested to the text, it is now Figure 2.

Thank you for your consideration of this manuscript.

Sincerely,

The authors.

Reviewer 2 Report

Good contribution. Requires minor revisions. See attached file.

Minor details. It never hurts to have it copyedited by a professional English copyeditor. However, this is not a must. In general, it is well written.

Author Response

Dear Reviewer, 

We are pleased to resubmit the manuscript of our paper entitled Decision Support Frameworks in Public Health Emergencies: A Systematic Review of Dynamic Models in Complex Contexts. We appreciate your recommendations, which considerably contributed to the improvement of the work. Please find a detailed description of the changes made below:

Point 1 - While the authors do a good job in discussing the relevance of the research topic, they do not indicate if there has been any previous review of decision support frameworks in PHE. Because, if so, they need to indicate what their new approach, limitations and contribution are to justify the need for a new systematic literature review. At the very least many literature reviews are justified in terms of time elapsed since the previous one and the need to catch up with new developments. 

Response 1 – We appreciate the feedback. We added two references to support the need for the work we are presenting (lines 239-252).

Point 2 - Related to the previous observation, while authors’ explanation of exclusion criteria E1 and E2 are self-explanatory based on the research questions, there are two exclusion criteria that deserve some explanation from the authors: E3 (exclude non-English written studies) and E4 (exclude research published before 2016). 

Point 2.1 - E3 may give as a limitation due to the lack of command of different languages and the difficulty of performing searches in many different languages but must be explicitly indicated.

Response 2.1 - Thank you for suggesting. The text of E3 was updated (lines 339-341), since we      understand that the main publications in this knowledge field are in English.

Point 2.2 - E4 requires more careful explanation. Why only the last 5 years? Aren’t PHE decision frameworks in use that have been developed prior to that? It is here that having explored previous literature reviews on the topic could be useful.  

Response 2.2 - As mentioned in Response 1, studies on decision-making in PHE are recent and a bibliometric analysis indicates an increase since 2014, associated with the intensification of health events, with 73.27% of the publications identified in the last 5 years, with an indication of growth in the coming years. The concern with PHE studies, especially on the measures of the PPRR model, continues to be a priority for dealing with future emergencies, such as the Bureau’s text of the WHO convention, agreement or other international instrument on pandemic prevention, preparedness and response (WHO CA+) [1]June 2023, that highlights the importance of each party shall, in accordance with applicable laws and supported by implementation plans, adopt policies, strategies and/or measures, that seek to integrate perspectives from public and private sectors and agencies, consistent with relevant tools or other international agreements. It shall strengthen and reinforce public health functions for: the continued provision of quality routine and essential health services during pandemics; sustaining and strengthening the capacities of the multidisciplinary workforce; collaborative surveillance, outbreak detection investigation and control, through interoperable early warning and alert systems; development of rehabilitation and post-pandemic health system recovery strategies; creating and strengthening public health institutions at national, regional and international, among others. In this sense, our article covers the last 7 years in a systematic literature review on aspects and configurations of decision support frameworks in PHE, which interprets the interface of management problems through the components of the H-EDRM (WHO, 2019) and the PPRR Model (Freitas et al., 2021; WHO, 2002), guided by questions to promote an interdisciplinary analysis.

Point 3 - Given that 36 is a manageable number of articles, Table 1 would be more useful if the columns would consist of an article identifier (from 1 to 24 or anyway you identified them in your study), a reference number  ([28], [29], etc) to locate the full cite in the reference list, authors’s names (first author et al), year, JCI, selected title and a brief description of what the article reports. Articles should be listed based on JCI from highest to lowest. This table could become a reference table for PHE decision support frameworks. 

Response 3 - We updated Table 1 with the information suggested (list, authors' names - first author et al -, year, JCI, selected title and a brief description of what the article reports) and identified the related H-EDRM and PPRR Model, as well as the central theme of the framework studied. This format allowed for a more comprehensive and analytical view, which will also support the reader in many combinations of analysis. As follow – Table 1:

AUTHORS

YEAR

JCI

SELECTED STUDIES - TITLE

HEDRM

PPRR
Mode

SUBJECT

Point 4 - There are only 17 articles listed in Table 2 while the dataset consists of 36 articles. I think authors may try to explain this in the last paragraph of p. 12 but I am not certain. It could be that since authors focused on identifying the different MCDA methods, they have only listed a single paper as a reference. It may be much easier to create a similar table to table 1. Use the paper#, reference number, first author et al, year, JCI, multicriteria method and application in the evaluated study.  

Response 4 – We appreciate the feedback. Table 2 has also been organized with the same elements (list, names of authors - first author et al -, year, JCI, selected title), and we have included APPLICATION OF THE MULTI-CRITERIA METHOD IN THE STUDY EVALUATED, which supports the analysis we are looking at in section 4 - Results. It is important to note that framework development incorporates decision support concepts, theories, and multicriteria methods, which address "multidimensional representation of problems with identification of critical information, comparisons of alternatives, and preference structures" (Gomes & Gomes, 2019). Although our study of decision support frameworks in PHE does not intend to exhaust the analysis of the limits and shortcomings of multi-criteria methods, which would imply considering that synergistic relationships should be strengthened between multi-criteria analysis, cognitive psychology, decision-maker behavior and decision support systems, our work identifies the methods and their applications cited in some studies. As follow – Table 2:

AUTHORS

YEAR

JCI

HEDRM

PPRR
Mode

MULTI-CRITERIA METHOD

APPLICATION IN THE EVALUATED STUDY

Point 5 - The appendix you link at in the first paragraph of Results is extremely useful and it is understandable it is too long to include it in the body of the paper; however, it would be necessary to link it to the paper references. For example, for the first paper, the author is Jing 2021 but it is laborious to find the whole paper reference in the manuscript reference list. I was able to locate Labib, 2021 from the appendix as [42] in the manuscript references.

Response 5 - Thank you for suggesting it. We have put the link to appendix i(https://bit.ly/47y1kBn) in Table 1 and Table 2, which allows the reader to access the file directly.    

[1] chrome-extension://efaidnbmnnnibpcajpcglclefindmkaj/https://apps.who.int/gb/inb/pdf_files/inb5/A_INB5_6-en.pdf

Thank you for your consideration of this manuscript.

Sincerely,

The authors.

Reviewer 3 Report

The paper is interesting. It has also some value. The authors have tried to make a comprehensive review. In fact, the authors have started very well; the first two sections- introduction and materials and methods are presented comprehensively. However, later on it seems the authors have lost some focus and consequently some incoherency is observed. The paper may be improved based on the following.

The objective of the study should be made clearer.

The section 3 does not bode well with the articulation. Although it is based on the literature review, and the authors have tried to answer the first of the research questions, it is not aligned well with the whole articulation leading to incoherency in reading. Either it should be located after the introduction section offering a theoretical background to the study or should be a part of section 4. Because it is equally important to that of the three questions answered in the results section.

The authors have answered the three important questions in more descriptive manner based on the meta-analyses. However, it would be more interesting and offer more Indepth and critical insights if the discussions are made under subthemes under each theme (question). So, the authors should answer the three questions under several subthemes under each theme.

One of the important and glaring aspect that is missing is uncertainty in the discussions. Any emergency situation is usually accompanied with uncertainty and entropy (Chaos). While the authors have to certain extent tried to consider Chaos (although not in a critical manner), uncertainty is totally ignored. Any strategy, framework or modeling will be incomplete without consideration for uncertainty. Discussions under subthemes might help to clearly elicit the issues and gaps. So, entropy and uncertainty may be included in the discussions. Moreover, the dynamicity of the models used and the complexity of the situations or contexts as observed from the review should be emphasized in the discussions.

The Tables and Graphical presentations have not been adequately cited in the text.

While in the section 4.1, reference to different countries have been made no such reference is made while answering the other two questions (section 4.2 and 4.3). Rather the discussions were made based on aspects. 

An explanation on why the literature review was limited to the publications between 2016 and 2023 is necessary.

I am not sure about the correctness of the total articles included in the review. The authors have mentioned that 36 articles have been found suitable and reviewed. Six articles were cited in the methodological approach. This leads to a total of 42 articles. However, the authors have cited 60 articles in the reference section. This should be verified as it forms an important aspect in the case of review-based articles.

The paper should be thoroughly edited, and citations should be made appropriately. A lot of error messages are found in the text.

The English language is acceptable. However, some editing is necessary. Some citations are either missing or wrong, as a results error message were found in the text.

Author Response

Dear Reviewer, 

We are pleased to resubmit the manuscript of our paper entitled Decision Support Frameworks in Public Health Emergencies: A Systematic Review of Dynamic Models in Complex Contexts. We appreciate your recommendations, which considerably contributed to the improvement of the work. Please find a detailed description of the changes made below:

Point 1 - The objective of the study should be made clearer.

Response 1 - We appreciate the feedback. Our intention is to analyze the frameworks in an approach that allows the reader, whether they are a health manager or professional or even researchers of decision-making models in PHE, to have an analysis of configurations, related aspects, model variables, as well as related gaps and challenges. Our work is not intended to exhaust the complexity of the analysis of each context in which the frameworks were inserted, but rather to analyze their importance for decision-making in PHE contexts. To make our Objective clearer, we updated it in the revised version (Abstract, lines 97-103).

What continuous to be a priority is to deal with future emergencies, such as mentioned by the Bureau’s text of the WHO convention, agreement or other international instrument on pandemic prevention, preparedness and response (WHO CA+) [1], June 2023, that highlight the importance of each party shall, in accordance with applicable laws and supported by implementation plans, adopt policies, strategies and/or measures, that seek to integrate perspectives from public and private sectors and agencies, consistent with relevant tools or other international agreements. It shall strengthen and reinforce public health functions for: the continued provision of quality routine and essential health services during pandemics; sustaining and strengthening the capacities of the multidisciplinary workforce; collaborative surveillance, outbreak detection investigation and control, through interoperable early warning and alert systems; development of rehabilitation and post-pandemic health system recovery strategies; creating and strengthening public health institutions at national, regional and international, among others.

Point 2 - The section 3 does not bode well with the articulation. Although it is based on the literature review, and the authors have tried to answer the first of the research questions, it is not aligned well with the whole articulation leading to incoherency in reading. Either it should be located after the introduction section offering a theoretical background to the study or should be a part of section 4. Because it is equally important to that of the three questions answered in the results section.

Response 2 - Section 3 has been changed to "Section 2 - Decision support frameworks in public health emergencies: concepts, complexities and managerial intervention capacity" and placed after the Introduction, focusing the main terms used in the article.

Point 3 - The authors have answered the three important questions in more descriptive manner based on the meta-analyses. However, it would be more interesting and offer more In-depth and critical insights if the discussions are made under subthemes under each theme (question). So, the authors should answer the three questions under several subthemes under each theme.

Response 3 - We appreciate the recommendation, it was essential to the reorganization of the text, allowing us to advance in the analyses by H-EDRM and PPRR Model, as well as to address the sub-themes listed in Figure 3 - Frameworks: scopes by H-EDRM. This improvement allowed for more in-depth and critical insights for the discussions and increased the number of pages from 24 to 30 (lines 439-797).

Point 4 - One of the important and glaring aspect that is missing is uncertainty in the discussions. Any emergency situation is usually accompanied with uncertainty and entropy (Chaos). While the authors have to certain extent tried to consider Chaos (although not in a critical manner), uncertainty is totally ignored. Any strategy, framework or modeling will be incomplete without consideration for uncertainty. Discussions under subthemes might help to clearly elicit the issues and gaps. So, entropy and uncertainty may be included in the discussions. Moreover, the dynamicity of the models used and the complexity of the situations or contexts as observed from the review should be emphasized in the discussions.

Response 4 - Reorganizing the text by H-EDRM and PPRR model, sub-themes, methods and gaps and analyzing their configurations, allowed us to identify aspects related to uncertainty. The process of generating uncertainty related to not knowing the complex behavior of a health event with an unknown pathogen, as we experienced with the impact of the Zika virus on neurological factors for pregnant women or even the expectation about the speed of the evolution of the SARS-CoV-2 infectivity rate, affects management systems in various dimensions, which we addressed in the 36 frameworks studied, as some examples that follows, described in Section 4.

Due to the difficulty of adjusting global strategies in PHE with measures for different realities and contexts, which requires constant adaptation.

  • Frameworks established by nations reflect the opinions and circumstances of their authors, the exchange of models between different country realities is not automatic, and it is vital to expand PHE research into the operational and ethical dimensions of models that address the dynamic complexity of management in a context (Khan et al., 2018)
  • The "one size fits all" approach strategies present governments with the difficulty of determining the spatial and temporal effects of different measures in relation to the characteristics of populations within the same country or territory, particularly those with greater economic and social vulnerability (Pan et al., 2022).
  • The models include characteristics of the nations' governance style in accordance with the degree of centralization of activities. The absence of command and the perception of PHE as a political game compromises the responses of governments, as in the United States during the Covid-19 pandemic, however the Chinese experience of more effective control of the pandemic is not easily adopted by other nations because their decision-making contexts differ (Jing, 2021).
  • The challenge to be overcome by the analysis frameworks is the development of assessment models with reproducible and validated processes, capable of incorporating available historical data or their absence for prevention and predictions, especially in new events or those extremely rare ones that generate uncertainties – the black swan (Keim, 2018).
  • The adjusting government strategies to contain SARS-CoV-2 transmission by analyzing the interactions and behaviors of agents integrates geographical factors, climatic factors, displacements and contacts between agents in public spaces, called resident in quarantine or confirmed case, the transportation agents or even the government agent which establishes policies to address the emergency. The restrictive measures for the population are incorporated according to the behavior of the agents and the dynamics of the contexts, without the prevalence of a one-size-fits-all approach. However, this situation produces the challenge for governments to know the impacts of different measures in space and time, given by the economic and social development of a region (Pan et al., 2022).
  • Knowing the performance of health systems in PHE allows an adaptive response for epidemics, which addresses analysis of context, intervention, process, performance and impact, and a route to performance analysis and optimization of interventions, but the studies show models focused on specific contexts, with low capacity to reformulate the intervention, frameworks with limited structures for reapplication in different contexts and iterative approach, missing settings of evaluation of the full cycle of an intervention (Warsame et al., 2020).

Due      to the lack of reliable data for projections, producing evidence and reducing uncertainties:

  • There is a lack of an 360° perspective on interventions in PHE settings, with evidence and best practices, considering different stakeholders in the decision-making process (Miglietta et al., 2021)
  • Despite the improvement of multicriteria decision support methods for health emergencies, the scoring adjustment of the attributes of analysis may be harmed in contexts of uncertainty, for dealing with strategies not previously tested and for more uncertain attributes that receive less weight in the analyses, compromising a broad view of the problem (Strong et al., 2021).
  • Modelling societal resilience in health emergencies integrating causation, barriers and lessons learned relies on hypothetical scenarios rather than probabilities and immediate problem-solving that is not always possible. Few studies have focused on frameworks for dynamically addressing the context of uncertainty, able to integrate accumulated management experiences from previous health crises to understand how contexts returned (recovery) to a state of normality (Labib, 2021)

Due to the number of variables that compromise the projection of measurements, the dynamic nature of the systems and uncertainties:

  • In problem analysis, planning for a PHE is a complicated system with interdependent variables (Khan et al., 2018). Thus, Multicriteria Decision Support Methods (MCDM) are used in multidimensional problem representation, identification of critical information, comparisons of alternatives and preference structures, being most effective in combined approach (Alkan & Kahraman, 2021; Asadi et al., 2022; Pegoraro et al., 2020). Despite the importance of mathematical models for scenario projection, the calibration of probabilities is more reliable for short time frames (two weeks), and degraded over the long term, according to the amount of trajectory uncertainty, intervention strategy, and decision quality in PHE (Funk et al., 2019).

Because of the different dynamics of society in a context of uncertainties that compromise the lives of everyone, especially vulnerable populations:

  • The framework is a decision tree that guides on the potential of the crisis, the establishment of recommendations, communications for intervention and verification of results, but the achievement of an adequate level of protection (prevention) for the population in public health emergencies, based on evidence, still lacks models that integrate the scientific, ethical and management dimensions, with social, infrastructure and logistical factors, for interventions that are appropriate to the context, especially for the least favored populations (McDonald et al., 2020).
  • Human resources can also be understood in the face of a PHE, by the integration of agents participating in the decision-making process (Kayman & Logar, 2016). The alignment of different agents in PHE decision-making request to overcome collective risk framing positioning, non-linear preferences, dependence on source conditions, and estimation of gains and losses of each actor or group of actors. Additionally, cognitive and complex linguistic information capture presents a challenge. The gaps point to the development of consensus models which integrate psychological factors, information quality, and decision speed. Thus, decision-maker hesitation and perception demand weights and ranking of alternatives in structured analysis, to mitigate their bounded rationality (Lv et al., 2022; Shi et al., 2022) and achieve an adequate deliberative process, by better integration of heuristics, biases and analysis of multiple variables (Kayman & Logar, 2016). It is difficulty to establish comprehensive, systematic, and standardized evaluation indexes on the real situation (Yang & Guo, 2022), when organizations and governments, individuals or groups act according to their own policies, resulting in compromised interoperability of systems and knowledge generation on common challenges, such as cross-border disease transmission (Neville et al., 2016).

Point 5 - The Tables and Graphical presentations have not been adequately cited in the text.

Response 5 – As previously mentioned, the recommendation to reorganize the text by H-EDRM and PPRR Model, allow us a better understanding of the process and to advance in a better analysis of each figure, which also covered the revision of Figure 3. In another example, we updated Table 1 with the information requested by reviewer 2 (list, authors' names - first author et al -, year, JCI, selected title and a brief description of what the article reports). We also identified the related H-EDRM and PPRR Model, as well as the central theme of the framework studied. This format allowed for a more comprehensive and analytical view, which will also support the reader in many combinations of analysis. As follow – Table 1:

AUTHORS

YEAR

JCI

SELECTED STUDIES - TITLE

HEDRM

PPRR
Mode

SUBJECT

Point 6 - While in the section 4.1, reference to different countries have been made no such reference is made while answering the other two questions (section 4.2 and 4.3). Rather the discussions were made based on aspects.

Response 6 – As mentioned before, the recommendation to reorganize the text by H-EDRM and PPRR Model, allowed us for more in-depth and critical insights for the discussions. We hope to have overcome this about the perception of countries and their challenges to build frameworks for decision-making in PHE, not country by country but following a perspective of development solutions according to each context.

Point 7- An explanation on why the literature review was limited to the publications between 2016 and 2023 is necessary.

Response 7 - We appreciate the feedback. We added two references to support the need for the work we are presenting (lines 239-252). The concern with PHE studies, especially on the measures of the PPRR model, continues to be a priority for dealing with future emergencies, such as the Bureau’s text of the WHO convention, agreement or other international instrument on pandemic prevention, preparedness and response (WHO CA+) [2]June 2023, that highlights the importance of each party shall, in accordance with applicable laws and supported by implementation plans, adopt policies, strategies and/or measures, that seek to integrate perspectives from public and private sectors and agencies, consistent with relevant tools or other international agreements. It shall strengthen and reinforce public health functions for: the continued provision of quality routine and essential health services during pandemics; sustaining and strengthening the capacities of the multidisciplinary workforce; collaborative surveillance, outbreak detection investigation and control, through interoperable early warning and alert systems; development of rehabilitation and post-pandemic health system recovery strategies; creating and strengthening public health institutions at national, regional and international, among others. In this sense, our article covers the last 7 years in a systematic literature review on aspects and configurations of decision support frameworks in PHE, which interprets the interface of management problems through the components of the H-EDRM (WHO, 2019) and the PPRR Model (Freitas et al., 2021; WHO, 2002), guided by questions to promote an interdisciplinary analysis.

Point 8 - I am not sure about the correctness of the total articles included in the review. The authors have mentioned that 36 articles have been found suitable and reviewed. Six articles were cited in the methodological approach. This leads to a total of 42 articles. However, the authors have cited 60 articles in the reference section. This should be verified as it forms an important aspect in the case of review-based articles.

Response 8 - As shown in Figure 2, the systematic review had 36 articles analyzed, the results of which can be found in section 4. The other titles are used for the theoretical basis of specific terms necessary for the development of our study, whether in the introduction, methodological approach, or theoretical basis. With the need for justification, including other works that addressed a systematic review on the topic, more works needed to be cited, such as the article by Hou et al., 2021 - Decades on emergency decision-making: a bibliometric analysis and literature review, 2021. This brought us to a total of 62 references, of which 36 were dedicated to the systematic review and 26 were complementary references.

Point 9 - The paper should be thoroughly edited, and citations should be made appropriately. A lot of error messages are found in the text.

Response 9 - We apologize for the errors, but in the article in PDF and word available on the MDPI platform we didn’t find the mistakes: https://susy.mdpi.com/user/manuscripts/displayFile/34aa75abe545fc1379350debacac81e7 ; https://susy.mdpi.com/user/manuscripts/displayFile/34aa75abe545fc1379350debacac81e7/latest_pdf . In any case, we have adjusted the references and sent the PDF version once again.      

Point 10 - Comments on the Quality of English Language. The English language is acceptable. However, some editing is necessary. Some citations are either missing or wrong, as a results error message were found in the text.

Response 10 - We apologize for the errors on the quality of English language. We revised the text to achieve the quality required for publication.

Thank you for your consideration of this manuscript.

Sincerely,

The authors.

[1] chrome-extension://efaidnbmnnnibpcajpcglclefindmkaj/https://apps.who.int/gb/inb/pdf_files/inb5/A_INB5_6-en.pdf

[2] chrome-extension://efaidnbmnnnibpcajpcglclefindmkaj/https://apps.who.int/gb/inb/pdf_files/inb5/A_INB5_6-en.pdf

Round 2

Reviewer 3 Report

The authors have addressed most of the major concerns. It may be acceoted with some minor editorial corrections.

The English language is acceptable.